# Determinants of Treatment Benefit and Post-Treatment Survival for Patients with Hepatocellular Carcinoma Enrolled in Second-Line Trials after the Failure of Sorafenib Treatment

**DOI:** 10.3390/jpm12101726

**Published:** 2022-10-17

**Authors:** Nicola Personeni, Tiziana Pressiani, Valentina Zanuso, Andrea Casadei-Gardini, Antonio D’Alessio, Martina Valgiusti, Vincenzo Dadduzio, Francesca Bergamo, Caterina Soldà, Mario Domenico Rizzato, Laura Giordano, Armando Santoro, Lorenza Rimassa

**Affiliations:** 1Department of Biomedical Sciences, Humanitas University, 20072 Milan, Italy; 2Medical Oncology and Hematology Unit, Humanitas Cancer Center, IRCCS Humanitas Research Hospital, 20089 Milan, Italy; 3Department of Oncology, IRCCS San Raffaele Scientific Institute, Vita-Salute San Raffaele University, 20132 Milan, Italy; 4Department of Surgery and Cancer, Faculty of Medicine, Imperial College London, London SW7 2AZ, UK; 5Department of Medical Oncology, IRCCS Istituto Romagnolo per lo Studio dei Tumori (IRST) “DinoAmadori”, 47014 Meldola, Italy; 6Oncology 1 Unit, Veneto Institute of Oncology, IOV, IRCCS, 35128 Padua, Italy; 7Department of Surgical, Oncological and Gastroenterological Sciences, University of Padua, 35122 Padua, Italy; 8Biostatistic Unit, Humanitas Cancer Center, IRCCS Humanitas Research Hospital, via Manzoni 56, 20089 Milan, Italy

**Keywords:** systemic therapy, second-line, immune checkpoint inhibitors, targeted agents, sorafenib, hepatocellular carcinoma

## Abstract

Second-line treatments are standard care for advanced hepatocellular carcinoma (HCC) patients with preserved liver function who are intolerant of or progress on first-line therapy. However, determinants of treatment benefit and post-treatment survival (PTS) remain unknown. HCC patients previously treated with sorafenib and enrolled in second-line clinical trials were pooled according to the investigational treatment received and the subsequent regulatory approval: approved targeted agents and immune checkpoint inhibitors (AT) or other agents (OT) not subsequently approved. Univariate and multivariate analyses using Cox proportional hazards models established relationships among treatments received, clinical variables, and overall survival (OS) or PTS. For 174 patients (80 AT; 94 OT) analyzed, baseline factors for longer OS in multivariate analysis were second-line AT, absence of both portal vein thrombosis and extrahepatic spread (EHS). Treatment with AT (versus OT) was associated with significantly longer OS among patients with EHS (p_interaction_ = 0.005) and patients with low neutrophil-to-lymphocyte ratio (NLR; p_interaction_ = 0.032). Median PTS was 4.0 months (95% CI 2.8–5.3). At second-line treatment discontinuation, alpha-fetoprotein (AFP) levels <400 ng/dl, albumin-bilirubin (ALBI) grade 1, and enrolment onto subsequent trials independently predicted longer PTS. Treatment with AT, PVT, and EHS were prognostic factors for OS, while AFP, ALBI grade and enrolment onto a third-line trial were prognostic for PTS. Presence of EHS and low NLR were predictors of greater OS benefit from AT.

## 1. Introduction

Systemic treatment is the only available option for hepatocellular carcinoma (HCC) patients with preserved liver function (Child–Pugh class A) presenting with advanced or intermediate HCC stages that are no longer suitable for locoregional procedures [1].

Since 2007, for such patients, the multikinase inhibitor (MKI) sorafenib has been worldwide considered the standard of care [2]. After a decade characterized by unsatisfactory results, novel targeted agents with prevalent antiangiogenic profiles have been approved in the context of first-line and second-line settings upon sorafenib discontinuation [3,4,5,6]. In addition, the advent of immune checkpoint inhibitors (ICIs) either as single agent or in combination has dramatically improved the therapeutic scenario, potentially resulting in a multitude of sequential treatment opportunities [7,8,9,10,11,12,13,14,15,16,17]. 

Today, the increased options for multiple lines of systemic therapies may improve the survival of patients affected by advanced HCC. However, on the other side, reliable factors that can predict treatment benefit as well as post-treatment survival remain unknown and represent an unmet clinical need. Even though new therapeutic strategies became available, sorafenib still is a valid first-line treatment option, especially for patients with absolute contraindications to ICIs [18]. 

Therefore, we aimed to investigate prognostic and predictive factors associated with outcomes in a cohort of sorafenib pre-treated patients who were fit for second-line treatments that were delivered in the context of clinical trials. In addition, as the therapeutic landscape of HCC is heading beyond the second line, we also investigated prognostic factors affecting survival after second-line treatment discontinuation. 

## 2. Materials and Methods

### 2.1. Patient Selection

From January 2012 to June 2018, all patients included in this cohort study received sorafenib as first-line treatment and a subsequent systemic agent in the context of clinical trials at three academic centers (IRCCS Humanitas Research Hospital, Rozzano (Milan); Istituto Oncologico Veneto-IRCCS, Padova; Istituto Scientifico Romagnolo per lo Studio e la Cura dei Tumori-IRCCS, Meldola) in Italy. They could have experienced disease progression under sorafenib, or they were deemed sorafenib-intolerant. Patients were further stratified according to the subsequent regulatory approval (if granted or not) of the investigational agent they had received during the trial. As a general rule, unless pre-specified, all study protocols for treatment of advanced HCC exclude patients with an Eastern Cooperative Oncology Group Performance Status (ECOG PS) ≥2 and a Child Pugh class B or C. Treatments could be discontinued due to (1) tumor progression; (2) unacceptable treatment toxicity, i.e., grade 2–4 AEs not responding to dose reductions and/or temporary interruption of treatment as per study protocol; (3) liver decompensation defined by the following clinical parameters: jaundice, ascites, gastrointestinal hemorrhage, or encephalopathy; (4) ECOG PS worsening, which was considered cancer progression. AEs were graded according to the National Cancer Institute Common Terminology Criteria for Adverse Events, version 4.03. This study was conducted in agreement with Good Clinical Practice guidelines, the ethical principles of the Declaration of Helsinki and local regulations. The protocol and its annexes were subject to review and approval by local Institutional Review Boards at each participating institution (ONC/OSS-01/2018). 

### 2.2. Outcome Measures

Patients’ data were retrospectively analyzed for baseline characteristics; second-line treatments received; and subsequent outcomes in terms of radiological response, OS, time-to-treatment failure (TTF), and post-treatment survival (PTS). 

Response was assessed according to RECIST v1.1. OS was measured as the date of enrollment on a second-line trial until death from any cause or date of last follow-up. TTF was measured from the first day of treatment on clinical trial until the patient came off study (for toxicity, disease progression, or death). The decision to discontinue study protocol was made by the treating physician based on patient’s history, AEs, and imaging studies. PTS was the time elapsed between the last day of treatment and death or last follow-up. 

We assessed potential predictors of survival including age, sex, Barcelona Clinic Liver Cancer (BCLC) stage, ECOG PS, previous surgery, sorafenib duration, reason for sorafenib discontinuation, time to start of second-line treatment, pattern of disease progression during first-line sorafenib, extra-hepatic spread (EHS), neutrophil-to-lymphocyte ratio (NLR), portal vein thrombosis (PVT), alpha-fetoprotein (AFP) levels, and liver function analyses. The albumin-bilirubin (ALBI) grade was calculated as previously reported [19].

### 2.3. Statistical Analysis

Demographical and clinical characteristics were summarized as number and percentages or as median and interquartile range (IQ range). The number of patients at the three participating institutions during the study period determined the sample size. Differences in distribution were estimated using the chi-square or the Fisher exact test (when appropriate) and the Wilcoxon’s *t*-test. Survival curves were generated using the Kaplan–Meier method. Differences between groups were evaluated using the log-rank test. A Cox proportional hazards regression model was used to calculate the hazard ratios (HR) and their 95% confidence intervals (95% CI). A multivariable model was built considering factors statistically significant in the univariate model which confirmed their effect. The relationship between baseline clinical characteristics and second-line treatment effect was evaluated using a Cox proportional hazards model with an interaction term. Statistical significance was set at *p* < 0.050. All reported *p* values were two-sided. All analyses were carried out with the SAS software v9.4.

## 3. Results

### 3.1. Baseline Patient Characteristics

Patients’ characteristics are summarized in Table 1. We included in this cohort study 174 patients with HCC who were treated with targeted agents approved for the second-line treatment of HCC; anti-programmed death receptor-1 (PD1) antibodies +/– anti-Cytotoxic T-Lymphocyte Antigen 4 (CTLA4) antibodies; anti-PD1 antibodies + targeted agents +/– anti-CTLA4 antibodies; other investigational treatments not reaching the approval for treatment of HCC. Targeted agents in this study include regorafenib, cabozantinib, and ramucirumab. Treatments not approved include cytotoxic chemotherapy, MET inhibitors, transforming growth factor-β receptor inhibitors, pleiotrophic pathway modifiers with immunomodulatory properties, MKI with antiangiogenic properties, cyclin-dependent kinases inhibitors, and placebo (delivered in the placebo arm of placebo-controlled trials). A slight majority of patients had low AFP levels as defined by the cut-off of 400ng/dL (n = 83 of 137 patients with AFP values available, 60.5%). The median time to sorafenib failure was 5.09 months (IQ range 2.8–9.5), and the median time from first-line sorafenib discontinuation until the first day of second-line treatment was 2.3 months (IQ range 1.4–4.9). 

### 3.2. Overall Survival from the Beginning of Second-Line Treatment

Radiological response to treatment was available for 162 patients (93.1%). Overall response rate (ORR) was 9.3% (*n* = 15), including partial responses. With a median follow-up of 36 months (IQ range 22.0–51.0), median OS was 9.7 months (95% CI 8.5–11.1). The survival of patients receiving placebo or agents not approved for HCC was similar, and the survival of patients receiving second-line targeted agents or anti-PD1 antibodies (alone or in combination) also tracked (Table 2). According to the subsequent regulatory approval, two investigational treatment categories were identified: approved treatments (AT, including targeted agents and anti-PD1 antibodies, alone or in combination) and other treatments (OT). Patients receiving AT had a significantly longer OS than patients receiving OT (HR 0.72, 95% CI 0.52–1.00, *p* = 0.048; Figure 1). In addition, the univariate analysis identified ECOG PS 0 (vs. 1), absence of PVT, absence of EHS, and prior surgery as baseline prognostic factors for longer OS (Table 3). Finally, a further subgroup analysis among patients with known nonviral etiology did not show a significant OS difference comparing targeted agents and ICIs-based treatment (median 11.1 months vs. 12.9 months, HR = 0.82, 95% CI 0.40–1.67; *p* = 0.589).

Multivariate analysis demonstrated that second-line treatment with AT, absence of both PVT and EHS were independent predictors of longer OS (Table 3). Significant interactions were detected between treatment and NLR as a continuous trait (*p* = 0.032), and between treatment and disease extent (*p* = 0.005, Figure 2A,B), suggesting that the benefit of AT over OT was significantly greater in patients with low NLR and EHS. 

### 3.3. Overall Survival from Start of Sorafenib

Patients from start of sorafenib had a median OS of 22.2 months (95% CI 18.9–24.6). Of note, for patients receiving OT, median OS was 21.9 months (95% CI 17.5–24.6), while for patients receiving AT median OS was 22.8 months (95% CI 18.4–28.1; *p* = 0.211). Compared with OT, survival rates with AT were higher, at 36 months after start of sorafenib through 60 months (Table 4). 

### 3.4. Time to Treatment Failure from Start of Second-Line Treatment

Median TTF for patients receiving OT was 3.6 months, and 4.3 months for patients receiving AT (HR, 0.69; 95% CI 0.51–0.94; *p* = 0.020). In particular, patients on treatment with targeted agents had a longer TTF than patients receiving OT (median 4.8 months vs. 3.6 months; HR = 0.67, 95% CI 0.46–0.98; *p* = 0.037), as did patients on treatment with anti-PD1 antibodies (median 4.1 months, HR = 0.72, 95% CI 0.49–1.06; *p* = 0.094). 

### 3.5. Post-Second-Line Treatment Survival

Median PTS was 4.0 months (95% CI 2.8–5.3). Apart from three patients still on treatment at their last follow-up, 125 (73.1%), 14 (8.2%), and 32 (18.7%) patients discontinued second-line treatments because of tumor progression, AEs, and liver failure or PS worsening, respectively (Table 5). In such patients median PTS was 5.3 (95% CI, 3.9–6.5), 5.0 (95% CI, 0.03–8.2), and 0.9 (95% CI, 0.5–1.4) months, respectively (*p* < 0.001). At the end of the second-line treatment, of 154 evaluable patients, 44 (28.5%) experienced a worsening of liver function (Child–Pugh Class B, 37 patients; Child–Pugh Class C, 7 patients). Interestingly, 28 patients (16.3%) with satisfactory ECOG PS and preserved liver function received a third-line treatment in a clinical trial. Besides clinical parameters (AFP levels, ALBI grade, NLR, reason for treatment discontinuation), univariate analysis identified radiological response to prior second-line treatment as a prognostic factor for PTS. Following multivariate analysis, low AFP levels (<400 ng/dl), enrolment onto third-line trial and ALBI grade 1 were independent prognostic factors for longer PTS (Table 6). 

## 4. Discussion

In this retrospective study, we investigated outcomes and prognostic factors for a cohort of patients with preserved liver function and an ECOG PS ≤ 1 being enrolled in second-line trials at three academic Italian Institutions after first-line sorafenib. Whereas according to published data nearly 25% of patients discontinue sorafenib due to liver function deterioration which precludes additional treatments, we sought to identify prognostic factors in the remainder who are eligible for clinical trial enrolment [20,21]. For OS, multivariate analyses indicate that treatment with OT and presence of PVT are negative prognostic factors. In addition, presence of EHS at the beginning of second-line treatments is both prognostic of shorter OS and predictive of increased AT benefit over OT. 

The survival outcomes are overall consistent with those reported across pivotal trials investigating targeted agents after first-line sorafenib [4,5,6] and those from a recent meta-analysis on the role of regorafenib as a valuable second-line treatment option after sorafenib [22]. Similarly, the survival figures with combinations of anti-PD1 antibodies plus targeted agents or anti-PD1 plus anti-CTLA4 are in line with the results of the most recent studies exploring different ICIs combinations [14,15,16,17]. Notably, in our cohort, the use of anti-PD1 antibodies + targeted agents +/– anti-CTLA4 antibodies resulted in numerically longer OS compared with approved agents (14.7 months vs. 10.9 months), even though this comparison, which was performed regardless of the etiology, relied on low numbers. Moreover, following the publication by Pfister et al suggesting a reduced efficacy of ICIs in patients with nonalcoholic steatohepatitis (NASH), we performed an exploratory survival analysis in a subset of patients with known nonviral etiology [23]. However, no significant survival difference was observed comparing ICIs-based treatment and targeted agents. Excepting the RESORCE trial, which enrolled only sorafenib-tolerant patients, the other second-line trials herein considered at each participating center had broadly overlapping inclusion criteria [4]. As such, all patients were assumed to represent a relatively homogeneous cohort, bearing preserved liver function (Child–Pugh A), ECOG PS ≤ 1, in addition to clinical and prognostic factors that allowed for subsequent trial enrolment. In contrast, patients who were given only frontline sorafenib were excluded from this analysis, as they represent a markedly different population bearing poor prognosis, driven by a deteriorating liver function [24,25]. The Child–Pugh status remains indeed a major prognostic determinant and it is a critical inclusion criterion for most clinical studies in HCC. 

During the past five years, the approval of numerous systemic treatment options has widened and changed the therapeutic scenario of advanced HCC. Even though the combination of atezolizumab plus bevacizumab is now considered the new first-line standard of care, sorafenib still represents a mainstay of advanced HCC treatment, since all second and further-line options have been evaluated in patients who were refractory or intolerant to sorafenib. Moreover, the optimal treatment sequencing after failure of the first line still represents a major topic, even considering the lack of determinants of treatment benefit and prognosis. 

We observed in univariate analysis a negative prognostic impact dictated by higher NLR. Interestingly, we also detected an increased benefit from AT over OT in patients with low NLR. These findings are consistent with prior analyses in both first- and second-line contexts [26,27,28]. In addition, in the immunotherapy era, a higher NLR has been also associated with worse survival in HCC patients treated with nivolumab monotherapy [29]. Similarly, lymphocyte-to-monocyte ratio (LMR) has been evaluated as inflammation-based biomarker. According to a meta-analysis, a higher LMR was associated with increased OS and disease-free survival (DFS) after liver resection [30]. Furthermore, both NLR and LMR were suggested to be independent prognostic factors for DFS after hepatectomy [31]. 

Additionally, we found a significant interaction between treatment and disease extent, that translated into more pronounced OS benefits from AT in patients with EHS. These results are in keeping with subgroup analyses of the RESORCE and CELESTIAL trials reporting greater OS benefits from regorafenib and cabozantinib in patients with EHS compared with patients with intrahepatic disease [4,5]. In contrast, the magnitude of benefit deriving from sorafenib over placebo was previously reported to be greater in patients without evidence of EHS [27]. In all, these findings may be helpful to gauge the benefit of approved second-line treatments in specific subgroups of patients.

Previous investigations showed that duration of first-line sorafenib does not affect time to progression nor OS after regorafenib [32,33]. However, available data are not consistent, as retrospective analyses from CELESTIAL suggest that OS in patients receiving cabozantinib or placebo could be related to prior sorafenib exposure [34].

In our experience, patients treated with sorafenib followed by AT had a median OS of 22.8 months (95% CI 18.4–28.1) from start of sorafenib, while the OS achieved sequencing sorafenib and OT was 21.9 months (95% CI 17.5–24.6). These findings suggest a better prognosis for patients potentially eligible for clinical trials [25]. Consistent with post hoc analyses of RESORCE and CELESTIAL, in patients receiving AT we estimated higher survival rates than those of patients on OT [33,34]. In particular, these were markedly diverging at 36 months from start of sorafenib, with a survival probability of 28% for patients treated with AT and 17% for patients treated with OT. This is in line with the hypothesis that survival outcomes are mostly driven by second-line treatments that are sequenced after sorafenib [24]. In this clinical scenario, OT represents a heterogenous group of treatments, including a placebo arm. However, results from previous negative trials showed similar outcomes for experimental arms and placebo arms [35,36,37,38]. Although exploratory in nature, the significant interactions we detected indicate that specific subgroups of patients, namely those with EHS and lower NLR, can derive greater survival benefit from AT over OT and deserve to be further investigated.

Most patients fail second-line treatments because of disease progression and AEs. Of note, across different second-line treatments considered, we observed similar rates of liver decompensation and PS worsening leading to treatment discontinuations. Regardless of the type of second-line agent, the overall PTS of 4 months we detected after second-line therapy align with the rates reported by Iavarone et al and by Shao et al. after first-line treatments [21,25]. To the best of our knowledge, the present study is also the first to investigate course and predictors of survival in patients who discontinue second-line AT or OT. To this end, we considered a previous model suggesting a strong prognostic impact linked to the reason of sorafenib discontinuation [21]. In this series, we observed a significantly longer OS in those patients who fail second-line treatment because of AEs or tumor progression, as compared to those who experience liver decompensation or PS worsening. Irrespective of treatment line, these latter patients still represent a hard-to-treat population better managed by best supportive care approaches. Eventually, low AFP levels, ALBI grade 1, and enrolment onto third-line trials (which recapitulates well-preserved liver function and absence of constitutional symptoms) emerge as independent prognostic factors for longer PTS. Despite the relatively small number of patients who were enrolled onto further studies, these findings clearly speak to the potential of third-line systemic treatments in selected patients. 

This study has several limitations related to its retrospective nature. Firstly, our study is based on data collected before the approval of ICI-based systemic treatment for advanced HCC, thus limiting the applicability of our results. Secondly, beyond the physicians’ expertise, some degree of heterogeneity in terms of disease management is possible across the tertiary referral centers involved. Thirdly, the pattern of disease progression after second-line treatment was not captured. Of note, the appearance of new extrahepatic lesions was reported to be linked to post-progression survival in post hoc analysis of REACH and REACH-2 studies [39]. Lastly, while we demonstrated that OS with placebo roughly aligns OS with agents not granted subsequent approval for HCC, among the latter, some could have had clinical activity and cannot be assumed to replace a placebo control.

## 5. Conclusions

This study reports on prognostic and predictive factors modulating the benefit of post-sorafenib AT over OT, as well as on the disease course once the second-line treatment has been discontinued. While reaffirming the prognostic impact of clinical parameters during second-line treatments and beyond, the current study indicates that patients with EHS and low NLR as subgroups experience greater OS benefits from AT. The fast-evolving therapeutic options in HCC in the last five years need to be addressed in similar studies, trying to define the best sequencing for patients. Our results constitute a base for a model to be update according to immunotherapy combinations use in first line. 

## Figures and Tables

**Figure 1 jpm-12-01726-f001:**
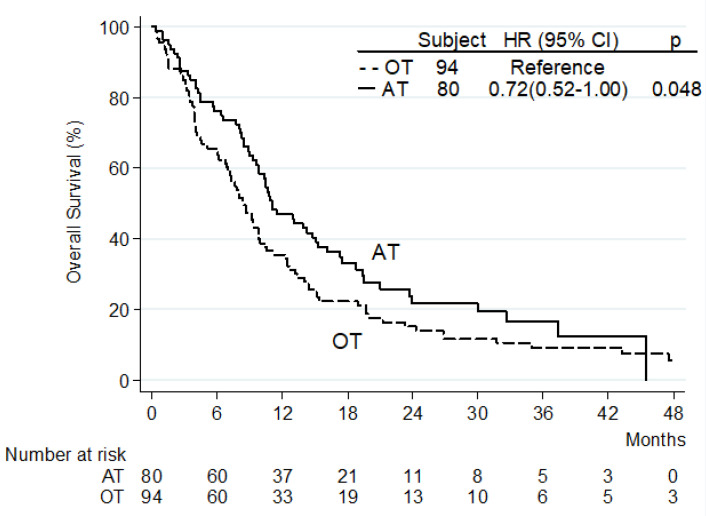
Overall survival by second-line treatment: approved treatments (AT) versus other treatments (OT).

**Figure 2 jpm-12-01726-f002:**
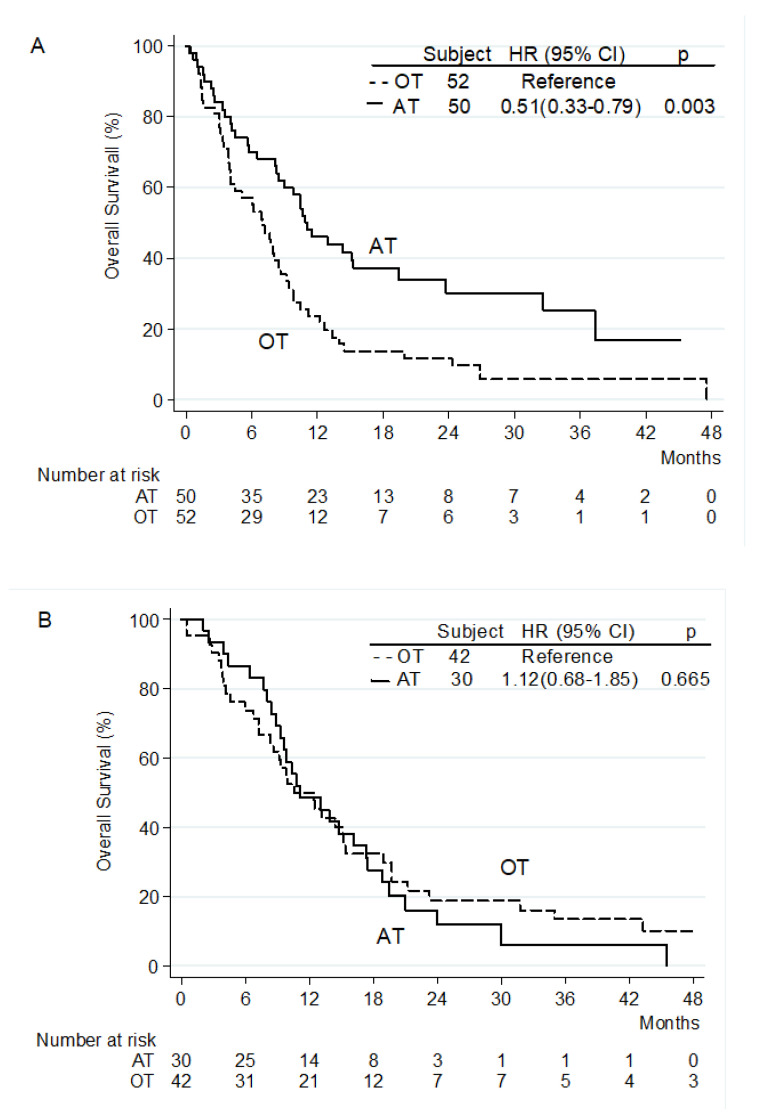
Analysis of disease extent as predictive factor for overall survival benefit from approved treatments. Extrahepatic spread (**A**) and intrahepatic disease (**B**), *p* value for disease extent treatment interaction = 0.005.

**Table 1 jpm-12-01726-t001:** Patient demographics and baseline characteristics.

Characteristics	*n* = 174 (%)
**Median Age**	
Years (range)	69 (24–85)
**Gender**	
Male	157 (90.2)
Female	17 (9.8)
**Diagnosis**	
Histology	117 (67.2)
AASLD criteria	57 (32.8)
**ECOG PS ^†^**	
0	103 (59.5)
1	70 (40.5)
**Barcelona Clinic Liver Cancer stage**	
B	37 (21.3)
C	137 (78.7)
**Etiology**	
Hepatitis C Infection	70 (40.2)
Hepatitis B Infection	23 (13.2)
Alcohol	34 (19.5)
Non-alcoholic fatty liver disease	15 (9.0)
Others	32 (18.3)
**Child–Pugh Class**	
A	172 (98.9)
B7	2 (1.1)
**Prior surgery**	
Yes	71 (40.9)
No	103 (59.1)
**Prior locoregional treatments**	
Yes	112 (64.4)
No	62 (35.6)
**Reason for sorafenib discontinuation**	
Disease progression	141 (81.0)
Adverse events	33 (19.0)
**Disease Extent**	
EHS	102 (58.6)
Intrahepatic only	72 (41.4)
**Portal Vein Thrombosis**	
Yes	50 (28.8)
No	124 (71.2)
**Median AFP**	
ng/dL (range)	86 (1–436300)
**Median neutrophils to lymphocyte ratio (range)**	3 (0–17)
**Second-line treatment**	
Targeted agents	40 (23.0)
ICI +/– targeted agents	40 (23.0)
Other treatments not approved for HCC	52 (29.9)
Placebo	42 (24.1)
**Reasons for second-line treatment discontinuation ^‡^**	
Disease progression	125 (73.1)
Adverse events	14 (8.1)
Liver failure or ECOG PS worsening	32 (18.8)
**Albumin-Bilirubin grade after second-line treatment ^§^**	
1	16 (22.9)
2	41 (58.5)
3	13 (18.6)

Abbreviations: AASLD, American Association for the Study of Liver Diseases; AFP, alpha-fetoprotein; ECOG PS, Eastern Cooperative Oncology Group performance status; EHS, extrahepatic spread; HCC, hepatocellular carcinoma; ICI, immune checkpoint inhibitors. ^†^ Missing data for one patient. ^‡^ Three patients were on treatment at last follow-up. ^§^ Missing data for 104 patients.

**Table 2 jpm-12-01726-t002:** Overall survival according to second-line treatment received.

Class of Drug	Median OS (Months)	HR (95% CI)	*p*-Value
Placebo	8.9	Reference	
Other treatments not approved for HCC	8.4	1.27 (0.83–1.96)	0.27
Targeted agents approved for HCC	10.9	0.89 (0.55–1.43)	0.63
Anti-PD1 antibodies +/– anti-CTLA4 antibodies	11.3	0.83 (0.45–1.51)	0.53
Anti-PD1 antibodies + targeted agents +/– anti-CTLA4 antibodies	14.7	0.68 (0.36–1.29)	0.24

Abbreviations: OS, overall survival; HR, hazard ratio; CI, confidence interval; HCC, hepatocellular; CTLA4, Cytotoxic T-Lymphocyte Antigen 4.

**Table 3 jpm-12-01726-t003:** Univariate and multivariate Cox analysis of factors associated with overall survival in 174 HCC patients undergoing second-line treatments.

	Univariate		Multivariate	
	HR (95% CI)	*p*-Value	HR (95% CI)	*p*-Value
**Age (continuous trait)**	1.00 (0.98–1.01)	0.58		
**Sex**				
Male/Female	1.55 (0.86–2.80)	0.15		
**ECOG PS**				
1/0	1.48 (1.06–2.07)	0.020		
**Barcelona Clinic Liver Cancer stage**				
B/C	0.76 (0.51–1.13)	0.17		
**HCC etiology**				
Non-viral/Viral (HBV or HCV-related)	0.88 (0.63–1.22)	0.42		
**Previous surgery**				
Yes/No	0.60 (0.43–0.84)	0.03		
**Sorafenib duration (continuous trait)**	1.00 (0.97–1.02)	0.67		
**Reason for sorafenib discontinuation**				
AEs/Disease Progression	0.69 (0.46–1.05)	0.08		
**Pattern of progression during first-line sorafenib**				
EHS/Intrahepatic	1.21 (0.84–1.74)	0.30		
**Time from sorafenib discontinuation to second-line start (continuous trait)**	1.01 (0.98–1.04)	0.56		
**Disease extent at the start of second-line treatment**				
EHS/Intrahepatic	1.21 (0.87–1.68)	0.26	2.14 (1.36–3.37)	0.001
**Portal vein thrombosis**				
Yes/No	1.91 (1.35–2.72)	<0.001	1.85 (1.28–2.69)	0.001
**AFP levels at start of second-line treatment (ng/dL)**				
≥400/<400	1.19 (0.82–1.73)	0.36		
**NLR (continuous trait)**				
High vs. low	1.51 (1.08–1.23)	<0.001		
**Second line treatment**	0.72 (0.52–1.00)	0.048	0.24 (0.12–0.47)	<0.001
AT/OT				

Abbreviations: AEs, adverse events; AFP, alpha-fetoprotein; AT, approved treatments; CI, confidence interval; ECOG PS, Eastern Cooperative Oncology Group performance status; EHS, extrahepatic spread; HBV, hepatitis B virus; HCC, hepatocellular carcinoma; HCV, hepatitis C virus; HR, Hazard ratio; NLR, neutrophil-to-lymphocyte ratio; OT, other treatments.

**Table 4 jpm-12-01726-t004:** Estimated survival rates from the start of sorafenib.

Survival Rate	Sorafenib → AT (*n* = 94)	Sorafenib → OT (*n* = 80)
6 months	100%	97%
12 months	81%	79%
24 months	46%	42%
36 months	29%	17%
48 months	19%	11%
60 months	12%	6%

Abbreviations: AT, approved treatments; OT, other treatments.

**Table 5 jpm-12-01726-t005:** Reasons for second-line treatment discontinuation.

Type of Treatment			
	Targeted Agents (*n* = 40)	Anti-PD1 Antibodies, Alone or in Combination (*n* = 40)	OT (*n* = 94)
Disease progression	25 (62%)	28 (70%)	72 (77%)
AEs	5 (13%)	2 (5%)	7 (7%)
Liver failure or ECOG PS worsening	10 (25%)	7 (18%)	15 (16%)
Ongoing treatment	-	3 (7%)	-

Abbreviations: AEs, adverse events; ECOG PS, Eastern Cooperative Oncology Group performance status; OT, other treatments.

**Table 6 jpm-12-01726-t006:** Univariate and multivariate Cox analysis of factors affecting post-treatment survival.

	Univariable		Multivariable	
	HR (95% CI)	*p*-Value	HR (95% CI)	*p*-Value
**Prior second-line treatment**	0.80 (0.58–1.12)	0.20		
AT/OT				
**Reason for second-line treatment discontinuation**				
Disease Progression/Liver Failure or PS worsening	0.38 (0.25–0.57)	<0.001		
AEs/Liver Failure or ECOG PS worsening	0.45 (0.23–0.87)	0.02		
**Enrolment onto third-line trial**				
Yes/No	0.31 (0.19–0.51)	<0.001	0.34 (0.15–0.78)	0.01
**AFP levels at second-line treatment discontinuation (ng/dL)**				
≥400/<400	1.67 (1.13–2.47)	0.010	2.01 (1.08–3.74)	0.029
**NLR at second-line treatment discontinuation** (continuous trait)				
	1.15 (1.08–1.22)	0.034		
**Radiological response (RECIST v 1.1 criteria) during prior second-line treatment**				
PD (or NA)/PR	2.67 (1.22–5.84)	0.014		
SD (or NA)/PR	1.52 (0.70–3.28)	0.293		
**ALBI grade at second-line treatment discontinuation**				
2/1	2.99 (1.36–6.54)	0.006	1.85 (0.81–4.21)	0.14
3/1	20.1 (7.4–54.55)	<0.001	7.53 (2.48–22.90)	<0.001

Abbreviations: AEs, adverse events; AFP, alpha-fetoprotein; AT, approved treatments; ECOG PS, Eastern Cooperative Oncology Group performance status; EHS, extrahepatic spread; HBV, hepatitis B virus; HCV, hepatitis C virus; HR, hazard ratio; NLR, neutrophil-to-lymphocyte ratio; OT, other treatments; SD, stable disease; PR, partial response; PD, progressive disease; NA, not available.

## Data Availability

Data are available on request from the authors.

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
