# Peer review of "Determinants of Treatment Benefit and Post-Treatment Survival for Patients with Hepatocellular Carcinoma Enrolled in Second-Line Trials after the Failure of Sorafenib Treatment"

_jpm, 2022, doi:10.3390/jpm12101726_

Round 1
Reviewer 1 Report
Very interesting and well written manuscript. I have only some minor comments
1) I would suggest to report quantitative data as median and interquartile range (instead of range).
2) Were Cox regression model assumptions verified? If so, how?
3) I appreciate the authors evaluated several markers such as ALBi or NLR. I would suggest to add some comments also on other markers such as LMR....
4) When speaking about regorafenib, i would cite the latest meta-analysis on this topic (PMID: 31877664)
Author Response
1) I would suggest to report quantitative data as median and interquartile range (instead of range).
As suggested by the reviewer, we amended the methods section as follows (line 159): Demographical and clinical characteristics were summarized as number and percentages or as median and interquartile range (IQ range).
Then, we reported the quantitative data according to medians and interquartile ranges in the following lines.
Lines 186 and 187-188: The median time to sorafenib failure was 5.09 months (IQ range 2.8-9.5) and the median time from first-line sorafenib discontinuation until the first day of second-line treatment was 2.3 months (IQ range 1.4-4.9).
Line 202: With a median follow-up of 36 months (IQ range 22.0-51.0).
2) Were Cox regression model assumptions verified? If so, how?
We verified the Cox regression model. In SAS using Assess statement and ph option, we performed the test for proportional assumption and the exploratory graphics for testing ph assumption. As shown in the Table below, no variable significantly violates the ph assumption (P values >.005). These data were not reported in the manuscript for the sake of clarity.
|
Variable |
P value |
|
Second line treatment |
0.1520 |
|
Portal Vein Thrombosis |
0.4930 |
|
Disease Extent |
0.4490 |
|
NLR |
0.8200 |
3) I appreciate the authors evaluated several markers such as ALBi or NLR. I would suggest to add some comments also on other markers such as LMR.
We thank the reviewer for the insightful statement. LMR has been evaluated as a prognostic inflammation-based biomarker and we agree about its importance in this context. Therefore, we edited the manuscript adding a specific paragraph in the discussion section, lines 338-342.
4) When speaking about regorafenib, I would cite the latest meta-analysis on this topic.
We thank the reviewer for the comment, and we added the suggested reference (discussion section, lines 303-304).
Reviewer 2 Report
This study demonstrated the survival outcome of second line treatments in patients who stopped sorafenib due to treatment failure. The main topic of this study is focused on comparing survival outcomes of approved treatments and non-approved treatments. However, this is not the main interest to the clinicians since non-approved treatments is not applied in most of the world for treatment on HCC. It only highlights the results of previously presented clinical trials.
Below, there are some more concerns regarding the manuscript.
1. Authors stated the aim of this study is to re-organize the treatment strategy for second-line treatment since the many first line treatment choices have been demonstrated after the introduction of lenvatinib. However, authors have not fulfilled the forementioned necessity of the study since it only demonstrated the treatment outcomes after the sorafenib failure which we all know owing to numerous former studies. To address this issue, author would have to redesign the study completely different and enroll the patients who experienced treatment failure with Lenvatinib or atezo-bevacizumab, but this cannot be addressed in this circumstance.
As for this reason, the introduction section should be reorganized to better explain the flow of logic of this article and focus the context only to the ‘factors associated survival outcome for those applying second line treatment who has conserved liver function after the sorafenib failure’.
2. Approved treatments vs non-approved treatments may not be the main interest for the readers. Rather comparing ICI with TKI is more intriguing
3. Subgroup analysis for non-viral etiology can also be presented. Survival outcome comparing ICI versus TKI in this subgroup could catch more attention of the readers due to known poor outcome of ICI in NASH.
4. To better clarify the context of the study, I suggest that authors change the title of the study to “Determinants of treatment benefit and post-treatment survival for patients with hepatocellular carcinoma enrolled in second-line trials after the failure of sorafenib treatment”
5. It seems that OT(‘other treatments’) include placebo arm, which makes the OT patients group extremely heterogenous. Since majority of AT drugs have been approved after showing superior results compared to the placebo-arm in terms of survival outcome, it is easy to predict that AT will outperform the OT group.
Author Response
This study demonstrated the survival outcome of second line treatments in patients who stopped sorafenib due to treatment failure. The main topic of this study is focused on comparing survival outcomes of approved treatments and non-approved treatments. However, this is not the main interest to the clinicians since non-approved treatments is not applied in most of the world for treatment on HCC. It only highlights the results of previously presented clinical trials.
We thank the reviewer for this general comment. However, the main interest of our study was to investigate prognostic and predictive factors associated with outcomes in sorafenib pre-treated patients who were fit for second-line treatments which were delivered in the context of clinical trials, as stated in the last paragraph of the introduction.
1) Authors stated the aim of this study is to re-organize the treatment strategy for second-line treatment since the many first line treatment choices have been demonstrated after the introduction of lenvatinib. However, authors have not fulfilled the aforementioned necessity of the study since it only demonstrated the treatment outcomes after the sorafenib failure which we all know owing to numerous former studies. To address this issue, author would have to redesign the study completely different and enroll the patients who experienced treatment failure with lenvatinib or atezo-bevacizumab, but this cannot be addressed in this circumstance.
As for this reason, the introduction section should be reorganized to better explain the flow of logic of this article and focus the context only to the ‘factors associated survival outcome for those applying second line treatment who has conserved liver function after the sorafenib failure’.
We thank the reviewer for the valuable consideration. We agree that the introduction could have been misleading, since we described the current treatment scenario, including novel therapeutic options and the challenging identification of the best sequence. Reliable factors that can predict treatment benefit as well as post-treatment survival remain unknown and represent an unmet clinical need. Since sorafenib has represented the mainstay of treatment for over a decade and still is a valuable treatment option, the aim of our work was to identify prognostic and predictive factors associated with outcomes in sorafenib pre-treated patients who undergo second-line treatments in the context of clinical trials. The introduction section has been completely amended.
2) Approved treatments vs non-approved treatments may not be the main interest for the readers. Rather comparing ICI with TKI is more intriguing.
We thank the reviewer for the suggestion. We specifically highlighted the OS benefit for ICI combinations strategies compared to approved targeted agents (discussion section, lines 307-310). However, this comparison relies on low numbers of patients. Moreover, it is important to point out that both ICIs alone or combined do not represent a standard second-line treatment in Europe (not approved by EMA) and are available only in clinical trials, thus not reflecting the proper real-world scenario.
3) Subgroup analysis for non-viral etiology can also be presented. Survival outcome comparing ICI versus TKI in this subgroup could catch more attention of the readers due to known poor outcome of ICI in NASH.
We thank the reviewer for this comment. We explored the possible impact of etiology on outcomes with a special focus on patients with non-viral etiology. Though some papers suggest a reduced efficacy of ICIs in patients with NASH, recent analyses (e.g., from the HIMALAYA trial) have shown inconsistent evidence and our data do not suggest significant differences comparing OS with TKIs or ICIs. However, our analysis was based on a very limited number of patients. We reported the data in the result section (lines 212-214) and added a comment in the discussion section (lines 311-314).
4) To better clarify the context of the study, I suggest that authors change the title of the study to “Determinants of treatment benefit and post-treatment survival for patients with hepatocellular carcinoma enrolled in second-line trials after the failure of sorafenib treatment”.
We thank the reviewer for the insightful suggestion. The title has been modified accordingly.
5) It seems that OT (‘other treatments’) include placebo arm, which makes the OT patients group extremely heterogenous. Since majority of AT drugs have been approved after showing superior results compared to the placebo-arm in terms of survival outcome, it is easy to predict that AT will outperform the OT group.
We concur with the reviewer regarding the superior outcomes with AT, as expected from pivotal trials that led to regulatory approvals. We also concur on the heterogeneity of the OT group. However, the OS outcomes in negative trials reported between 2012 and 2018 tend to be similar, in the experimental arm and in the placebo arm. These findings rule out any prognostic effect carried by the treatment arm.
Primary aim of this analysis was to describe clinical factors that may help to maximize treatment benefit. Although exploratory in nature, the significant interactions we detected indicate that particular subgroups of patients -namely those with EHS and lower NLR- can derive greater survival benefit from AT over OT. Whereas the PFS data reported in pivotal trials suggest similar benefits from AT in patients with or without EHS, we believe that additional prognostic factors are determinants of OS in this latter subgroup.
We added a paragraph in the discussion section (lines 364-369).
Lastly, we confirm that we have reviewed English in our manuscript.
Round 2
Reviewer 2 Report
I am satisfied with the reply from authors